🔓 | **Open Peer Review** | Public Health | Research Article

# Prevalence and genetic diversity of intestinal parasites in Xilingol League herbivores, Inner Mongolia, Northern China

Ziran Mo,[1] Jingwei Quan,[1] Jiahao Dao,[1] Xiaoping Luo,[2] Penglong Wang,[2] Yurun Miao,[1,3] Lei Xiu,[1,4] Gaowa Gong,[2] Jian Li,[1] Wenbin Yang,[1,4] Junyan Li,[2] Wei Hu[1,4,5,6]

**ABSTRACT** Intestinal parasites are common infectious agents that substantially impact the health and productivity of livestock, with several species posing zoonotic risks. In this study, 205 fecal samples were collected from cattle, sheep, and goats in Xilingol League. Microscopic examination was performed to detect *Eimeria* spp., nematodes, and trematodes, whereas nested polymerase chain reaction assays were used to identify *Giardia duodenalis*, *Cryptosporidium* spp., and *Enterocytozoon bieneusi*. The results showed that the overall prevalence of intestinal parasites was 80.5% (165/205) among herbivores, with a co-infection rate of 47.8% (98/205). Across host species, the prevalence rates were 92.8% (126/139) in sheep, 75.0% (12/16) in goats, and 52.0% (26/50) in cattle. The prevalences of *Eimeria* spp., nematodes, trematodes, *G. duodenalis*, *Cryptosporidium* spp., and *E. bieneusi* were 59.5%, 16.1%, 33.7%, 25.9%, 7.8%, and 4.9%, respectively. Genetic characterization further revealed distinct dominant lineages, including *G. duodenalis* (assemblages B and E), *Cryptosporidium* spp. (*C. andersoni* and *C. xiaoi*), and *E. bieneusi* (genotypes J and BEB6). Furthermore, infection with *Eimeria* spp. was significantly associated with an increased likelihood of co-infection with other intestinal parasites. Together, these findings provide a comprehensive overview of intestinal parasite co-infections among herbivores in Xilingol League and highlight pronounced interspecies differences and regional variation in parasite distribution.

**IMPORTANCE** Intestinal parasitic infections remain a major constraint for animal health and productivity, especially in large-scale grassland livestock systems. Despite their economic and zoonotic implications, comprehensive assessments of intestinal parasite co-infection patterns in these ecosystems have been lacking. This study addresses this gap by characterizing the prevalence, diversity, and genetic makeup of key intestinal parasites in Xilingol League. The detection of frequent co-infections and predominant zoonotic genotypes underscores the importance of integrated parasite management strategies. Furthermore, infection with *Eimeria* spp. was significantly associated with co-infections involving other intestinal parasites, suggesting potential interactions that require further investigation. These insights are critical for informing region-specific parasite control measures, improving veterinary health outcomes, and reducing potential public health threats at the livestock-human interface.

**KEYWORDS** intestinal parasite, Xilingol, epidemiology, genotype, haplotype

Intestinal parasitic infections are a major global public health challenge, affecting over 2 billion people, particularly in low-resource regions lacking adequate sanitation (1, 2). These infections, caused by protozoa (e.g., *Cryptosporidium* spp., *Giardia duodenalis*, *Enterocytozoon bieneusi*, *Eimeria* spp.) and helminths (e.g., *Ascaris* spp., *Taenia* spp., *Trichuris* spp.), are primarily transmitted via the fecal-oral route and are associated with gastrointestinal disorders, malnutrition, and potentially fatal outcomes in

**Peer Reviewer** Si-Yang Huang, Yangzhou University, Yangzhou, China

Address correspondence to Wenbin Yang, yangwb@imu.edu.cn, Junyan Li, lijyjsc@163.com, or Wei Hu, huw@imu.edu.cn.

Ziran Mo and Jingwei Quan contributed equally to this article. The author order was determined in order of decreasing seniority.

The authors declare no conflict of interest.

See the funding table on p. 12.

immunocompromised individuals (3–7). In livestock, intestinal parasites impair growth performance, reduce product quality, and impose substantial economic losses, while infected animals serve as reservoirs that contaminate the environment and facilitate zoonotic transmission (8–11). Therefore, systematic epidemiological investigations of intestinal parasites in livestock are essential to elucidate infection dynamics, transmission patterns, and population genetic diversity, thereby supporting sustainable animal husbandry, ensuring food safety, and protecting public health.

Intestinal parasitic infections are widespread in livestock worldwide, with infection prevalence often exceeding 70% in some regions (12–17). Inner Mongolia, particularly the Xilingol League, represents one of China's most important pastoral areas, supporting more than 10 million grazing animals across approximately 180,000 km$^2$ of grassland (18–20). While *Cryptosporidium* spp. and *G. duodenalis* infections in cattle have been reported (21, 22), comprehensive epidemiological data on intestinal parasites, especially multi-species infections, remain limited in this region. To date, only one study has documented nematode infections in local sheep (23), highlighting the lack of baseline information on intestinal parasite diversity and transmission risks in Xilingol's grazing systems.

Given the region's open grazing environment and constant exposure of livestock to environmental sources of infection, systematic surveillance is essential for assessing infection pressure and transmission potential. Intestinal parasitic infections frequently occur as co-infections involving multiple parasite species, which can exacerbate pathological outcomes and increase transmission potential (24, 25). However, most existing regional surveys have focused on single-parasite detection or overall prevalence, providing limited insight into co-infection dynamics and interspecies interactions. Therefore, comprehensive and systematic investigations are needed to clarify the occurrence, interactions, and epidemiological impacts of intestinal parasite co-infections in livestock populations.

To address this knowledge gap, a systematic investigation of herbivores from three major livestock areas in the Xilingol League was conducted, using both microscopic examination and nested PCR to detect *Eimeria* spp., nematodes, trematodes, *G. duodenalis*, and *Cryptosporidium* spp., and *E. bieneusi*. This study establishes baseline data on parasite prevalence and genetic diversity in the region and provides essential evidence for developing targeted control strategies to improve livestock productivity, ensure food safety, and reduce zoonotic risks.

## RESULTS

### Epidemiological characteristics of *Eimeria* spp., nematodes, and trematodes

Microscopic examination was performed to investigate intestinal parasites in livestock across multiple sites within the Xilingol League. Among the detected parasites, *Eimeria* spp. oocysts were the most frequently observed. The overall infection rate of *Eimeria* spp. was 59.5% (122/205) (Fig. 1; Table 1), with variation observed among host species and sampling regions. Across regions, infection rates were highest in East Ujimqin County (65.0%, 91/140) and Xilinhot (67.4%, 29/43), both significantly higher than in Sonid Left Banner (9.1%, 2/22) (DW vs SZ: odds ratio [OR] = 18.57, 95% confidence interval [CI]: 4.41–81.86, $P < 0.05$; BY vs SZ: OR = 20.71, 95% CI: 4.34–94.79, $P < 0.05$). Host-specific analysis showed that sheep had a higher infection rate (78.4%) compared to goats (50.0%; OR = 3.63, 95% CI: 1.20–10.91, $P < 0.05$) and cattle (10.0%; OR = 32.7, 95% CI: 11.82–79.85, $P < 0.05$). Regional variation was also evident within the same host species, with infection in sheep from East Ujimqin County (87.9%) significantly higher than in Xilinhot (67.4%) (OR = 3.51, 95% CI: 1.37–8.58, $P < 0.05$). Overall, *Eimeria* spp. infection exhibited distinct regional and host-associated differences, with the highest prevalence recorded in sheep and in samples collected from East Ujimqin County and Xilinhot.

Microscopic examination was performed to investigate nematode and trematode infections in livestock across multiple sites within this region. Nematode and trematode eggs were detected in 16.1% (33/205) and 33.7% (69/205) of fecal samples, respectively

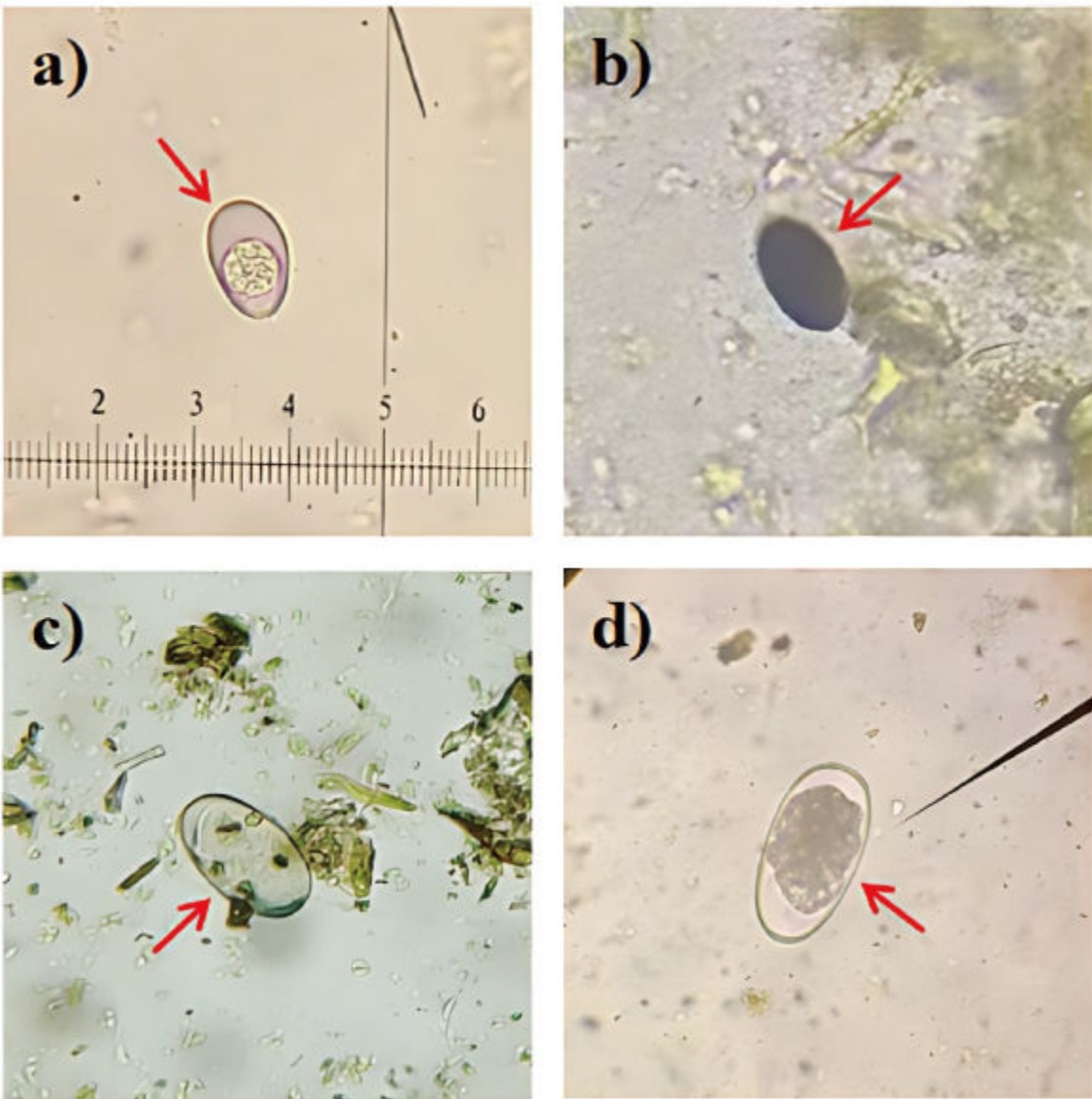

**FIG 1** Oocysts or eggs of the infected gastrointestinal parasites ([a] *Eimeria* spp. oocyst; [b] *Dicrocoelium* spp. egg; [c] *Paramphistomum* spp. egg; [d] *Haemonchus* spp. egg).

(Fig. 1; Table 1). The prevalence of nematode infections showed considerable regional variation, with the highest infection rate in Xilinhot (46.5%), significantly higher than that in East Ujimqin County (7.1%; OR = 10.29, 95% CI: 4.361–25.49, $P < 0.05$) and Sonid Left Banner (13.6%; OR = 12.03, 95% CI: 2.72–43.29, $P < 0.05$). By host species, the prevalence of nematodes was higher in sheep (28.8%) than in goats (6.3%) and cattle (0%), although no statistically significant difference was detected among hosts. Within the sheep population, infection in Xilinhot was significantly higher than in East Ujimqin County (OR = 6.41, 95% CI: 2.66–16.20, $P < 0.05$). The prevalence of trematodes was relatively consistent among regions, with East Ujimqin County (36.4%), Sonid Left Banner (31.8%), and Xilinhot (25.6%), and no significant regional variation was detected. Among the host species, cattle exhibited the highest prevalence (42.0%), followed by goats (31.3%) and sheep (30.9%), though these differences were not statistically significant. Overall, nematode infections exhibited pronounced regional and host-related variation,

**TABLE 1** Occurrence of *Eimeria* spp., *Haemonchus* spp., *Fasciola* spp., *Cryptosporidium* spp., *E. bieneusi*, and *G. duodenalis* in different regions of XilinGol League, China

| Region | Host | No. specimens | No. positive for *Eimeria* spp. (%) | No. positive for *Haemonchus* spp. (%) | No. positive for *Fasciola* spp. (%) | No. positive for *Cryptosporidium* spp. (%) | *Cryptosporidium* species (no.) | No. positive for *E. bieneusi* (%) | *E. bieneusi* genotype (no.) | No. positive for *Giardia* | Assemblage (no.) |
|---|---|---|---|---|---|---|---|---|---|---|---|
| Sonid Left | Cattle | 12 | 2 (16.7) | 0 (0) | 4 (33.3) | 1 (8.3) | *C. andersoni* (1) | 0 (0) | – | 1 (8.3) | B (1) |
| | Goat | 5 | 0 (0) | 1 (20.0) | 1 (20.0) | 2 (40.0) | *C. xiaoi* (1) | 0 (0) | – | 1 (20.0) | B (1) |
| | Sheep | 5 | 0 (0) | 2 (40.0) | 2 (40.0) | 2 (40.0) | *C. xiaoi* (2) | 0 (0) | – | 2 (40.0) | B (2) |
| Xilinhot | Sheep | 43 | 29 (67.4) | 20 (46.5) | 11 (25.6) | 7 (16.3) | *C. xiaoi* (7) | 0 (0) | – | 6 (14.0) | E (1), B (5) |
| East Ujimqin | Cattle | 38 | 3 (7.9) | 0 (0) | 17 (44.7) | 1 (2.6) | *C. andersoni* (1) | 3 (7.9) | J (2), BEB6 (1) | 3 (7.9) | B (3) |
| | Goat | 11 | 8 (72.7) | 0 (0) | 4 (36.4) | 0 (0) | –[a] | 2 (18.2) | BEB6 (2) | 6 (54.6) | B (6) |
| | Sheep | 91 | 80 (87.9) | 10 (11.0) | 30 (33.0) | 3 (3.3) | *C. xiaoi* (3) | 5 (5.5) | BEB6 (5) | 34 (37.4) | B (34) |
| Subtotal | | 205 | 122 (59.5) | 33 (16.1) | 69 (33.7) | 16 (7.8) | *C. andersoni* (2), *C. xiaoi* (14) | 10 (4.9) | J (2), BEB6 (8) | 53(25.9) | B (52), E (1) |

[a]–, not detected.

whereas trematode infections remained relatively consistent across regions and host species.

Collectively, gastrointestinal parasite infections in livestock from Xilingol League demonstrated distinct epidemiological patterns, with overall prevalence rates of 59.5% for *Eimeria* spp., 33.7% for trematodes, and 16.1% for nematodes.

## Prevalence, phylogenetic analysis, and haplotype characterization of *G. duodenalis*

Molecular detection was performed to investigate *G. duodenalis* infection and genetic diversity in livestock across multiple sites within the Xilingol League. The overall prevalence of *G. duodenalis* was 25.9% (53/205) (Table 1). Across regions, infection rates were highest in East Ujimqin County (30.7%), followed by Sonid Left Banner (18.2%) and Xilinhot (14.0%). Host-specific analysis showed that goats (54.6%) and sheep (37.4%) had a significantly higher infection risk than cattle (7.9%) (goat vs cattle: OR = 8.94, 95% CI: 1.99–30.49, $P < 0.05$; sheep vs cattle: OR = 4.98, 95% CI: 1.81–13.54, $P < 0.05$). Within the sheep population, the infection rate in East Ujimqin County (37.4%) was significantly higher than in Xilinhot (14.0%) (OR = 3.69, 95% CI: 1.49–9.30, $P < 0.05$).

Phylogenetic analysis based on the β-giardin (bg) gene locus demonstrated that the *G. duodenalis* sequences obtained in this study clustered into distinct clades corresponding to assemblages B and E, which showed high sequence similarity to strains previously reported from Chifeng and other areas of Inner Mongolia (Fig. 2A). Haplotype network analysis further revealed regional variation in haplotype composition. The predominant haplotype, Hap_3 (75.6%), was widely distributed across all three regions (Fig. 2B). East Ujimqin County exhibited greater haplotype diversity than Xilinhot and Sonid Left Banner, with three distinct haplotypes (Hap_1, Hap_2, and Hap_3) identified in this region, whereas only Hap_3 was detected in Xilinhot and Sonid Left Banner (Fig. 2B).

Overall, *G. duodenalis* infection in livestock from the Xilingol League exhibited distinct regional and host-associated variation, with assemblages B and E as the predominant genotypes and Hap_3 as the most prevalent haplotype across the study area.

## Prevalence, phylogenetic analysis, and haplotype characterization of *Cryptosporidium* spp.

Nested PCR detected *Cryptosporidium* spp. infections in livestock across multiple sites within the Xilingol League, with *C. andersoni* and *C. xiaoi* identified as the detected species. The overall prevalence of *Cryptosporidium* spp. was 7.8% (16/205) in cattle and

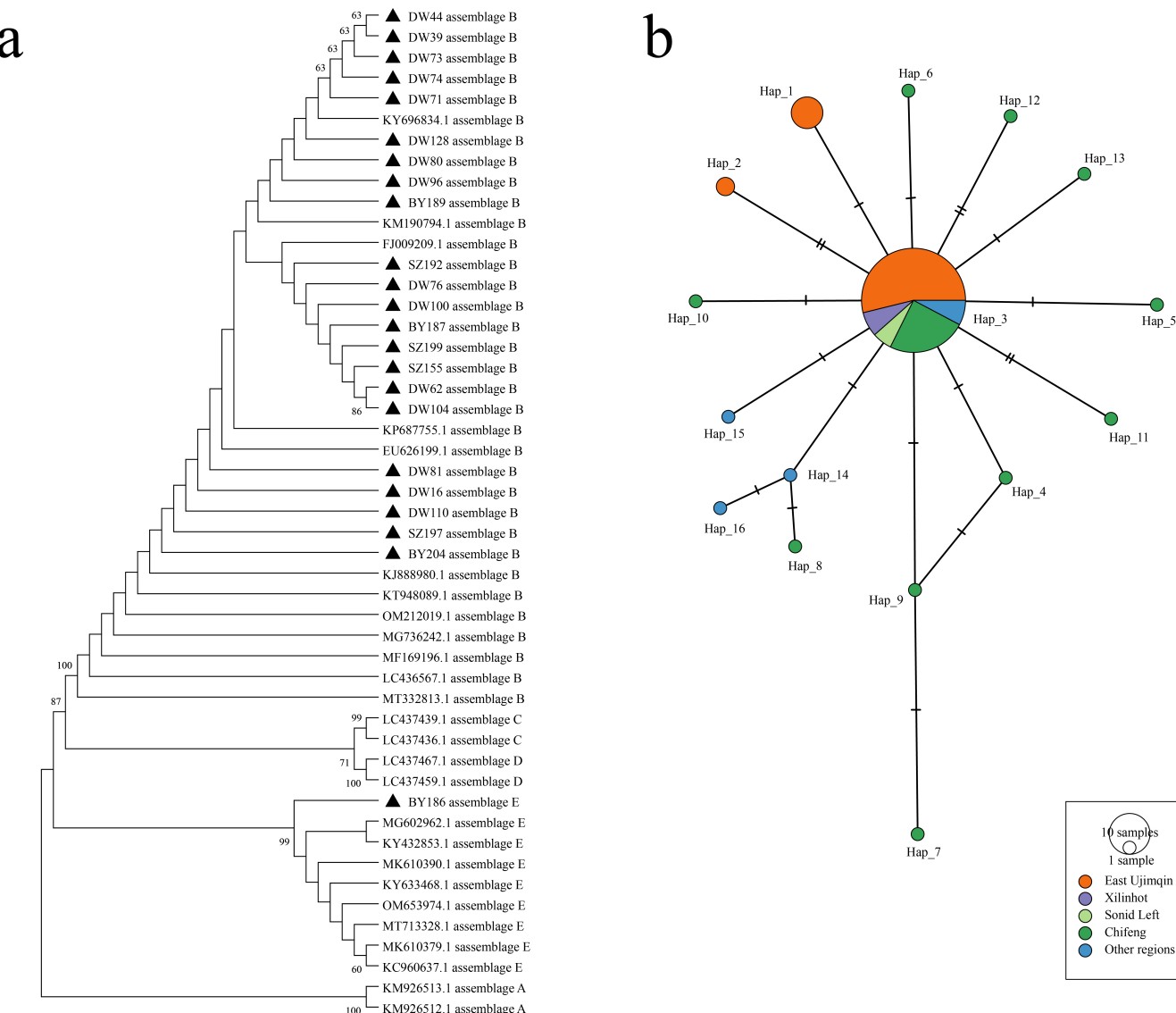

**FIG 2** Phylogenetic tree and haplotype analysis of *G. duodenalis*. (a) Phylogenetic tree of the *G. duodenalis bg* gene. (b) Haplotype analysis results of *G. duodenalis* sequences detected in this study and published *G. duodenalis* sequences. Different colors represent different groups, including DW, BY, SZ, Chifeng, and other regions, and the size of the circles represents the number of haplotypes.

sheep (Table 1). Across regions, East Ujimqin County had the lowest infection rate (2.9%), which was significantly lower than that in Sonid Left Banner (22.7%, OR = 10.00, 95% CI: 2.55–34.26, $P < 0.05$) and Xilinhot (16.3%, OR = 6.61, 95% CI: 1.92–20.80, $P < 0.05$). Within the sheep population, the infection rate in East Ujimqin County (5.5%) was significantly lower than that in Xilinhot (16.3%, OR = 5.70, 95% CI: 1.43–20. 87, $P < 0.05$) and Sonid Left Banner (40.0%, OR = 19.56, 95% CI: 2.48–119.60, $P < 0.05$). No significant regional differences were observed in the infection rates among cattle.

Phylogenetic analysis based on the small subunit ribosomal RNA (SSU rRNA) gene locus revealed that *C. andersoni* and *C. xiaoi* clustered into two distinct clades (Fig. 3A). Haplotype network analysis identified eight haplotypes among the sequences obtained in this study. The predominant haplotypes were Hap_3 (38.4%), Hap_7 (23.1%), and Hap_8 (30.8%). Hap_3 and Hap_8 were identified in both Xilinhot and Sonid Left Banner, whereas only Hap_7 was detected in East Ujimqin County (Fig. 3B).

Overall, *C. xiaoi* was the most frequently detected *Cryptosporidium* species in livestock from the Xilingol League, with Hap_3 and Hap_8 identified as the predominant

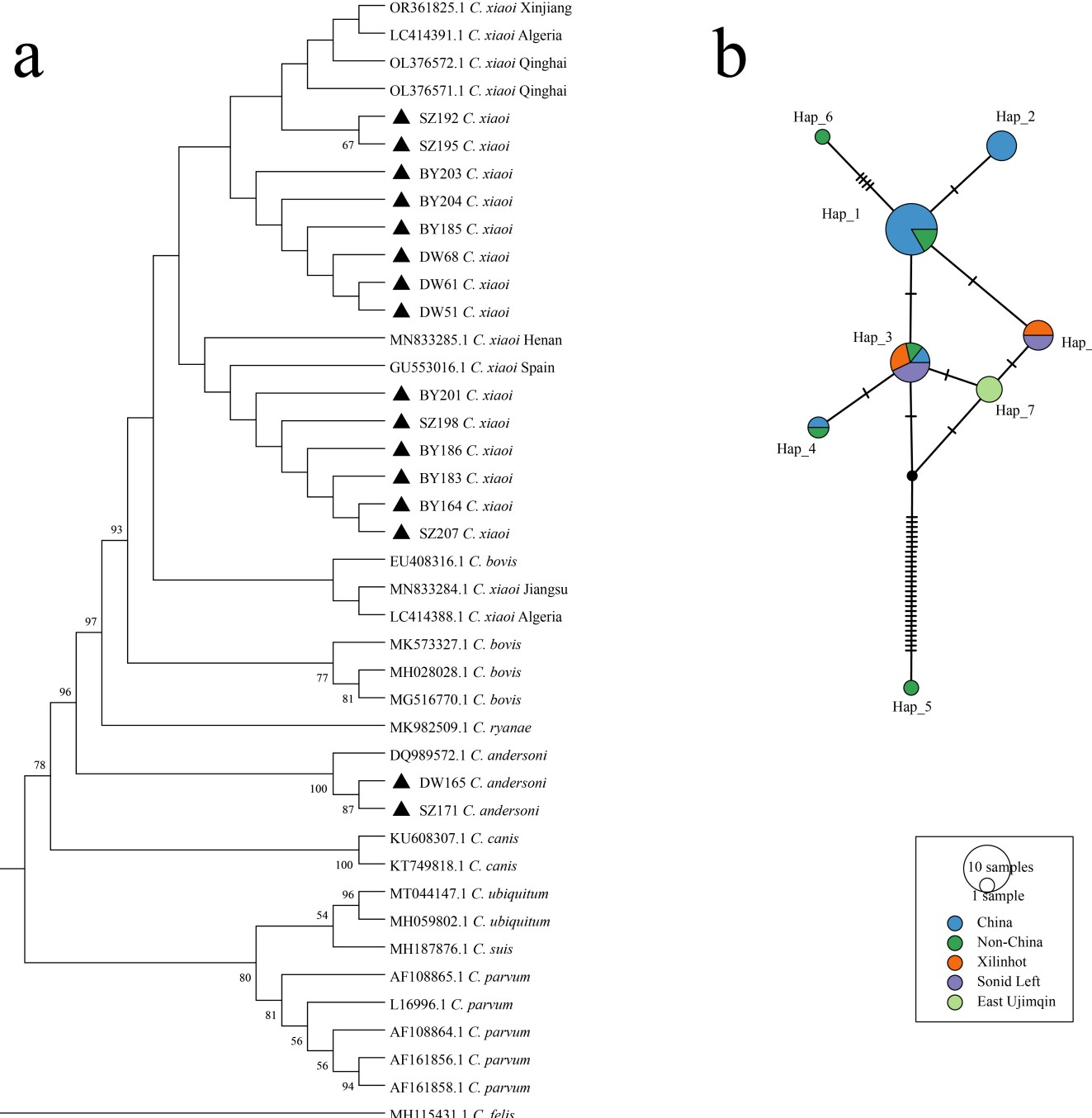

**FIG 3** Phylogenetic tree and haplotype analysis of *Cryptosporidium* spp. (a) Phylogenetic tree of the *Cryptosporidium* spp. *SSU rRNA* gene. (b) Haplotype analysis results of *C. xiaoi* sequences detected in this study and published *C. xiaoi* sequences. Different colors represent different groups, including DW, BY, SZ, China, and non-China, and the size of the circles represents the number of haplotypes.

haplotypes. Marked regional variation was observed in infection rates and haplotype distribution among host populations.

## Prevalence, phylogenetic analysis, and haplotype characterization of *E. bieneusi*

Sequence-based molecular analysis revealed *E. bieneusi* infections in livestock across multiple sites within the Xilingol League. The overall prevalence of *E. bieneusi* in cattle

and sheep was 4.9% (10/205) (Table 1). Positive samples were detected in East Ujimqin County, where the infection rate reached 7.2% (10/140); no infections were identified in Sonid Left Banner or Xilinhot. The infection rates among different host species were as follows: cattle, 7.9% (3/38); goats, 18.2% (2/11); and sheep, 5.5% (5/91). Although the infection rate varied among host species, the differences were not statistically significant.

Phylogenetic analysis based on internal transcribed spacer (ITS) gene sequences revealed that *E. bieneusi* genotypes J and BEB6 clustered within group 2, a lineage predominantly found in ruminants (Fig. 4A). Haplotype network analysis incorporating sequences from the GenBank database identified 22 haplotypes in total. Among these, only 2 were detected in the present study: Hap_9, which was primarily found in goats and sheep from East Ujimqin County, and Hap_2, which was detected exclusively in cattle (Fig. 4B). Hap_9 was the predominant haplotype in East Ujimqin County, accounting for 80% of the detected sequences, and exhibited distinct regional and host-associated distribution patterns.

Overall, *E. bieneusi* infections in livestock from the Xilingol League were confined to East Ujimqin County, where genotype diversity was limited to two group 2 genotypes (J and BEB6) represented by distinct host-associated haplotypes.

## Overall prevalence of gastrointestinal parasites in cattle and sheep from Xilingol League

Epidemiological analysis identified widespread intestinal parasite infections and frequent co-infections among cattle and sheep in the Xilingol League. Statistical analysis showed that the overall prevalence of intestinal parasites in cattle and sheep was 80.5% (165/205), with at least one of the following parasites detected in each positive sample: *Eimeria* spp., nematodes, trematodes, *Cryptosporidium* spp., *E. bieneusi*, and *G. duodenalis* (Table S1). Significant regional differences in infection rates were observed. The infection rate in livestock from Sonid Left Banner (50.0%, 11/22) was significantly lower than that in East Ujimqin County (85.0%, 119/140; OR = 0.18, 95% CI: 0. 07–0.47, $P < 0.05$) and Xilinhot (79.1%, 34/43; OR = 0.26, 95% CI: 0.10–0.80, $P < 0.05$). Host species and regional factors both influenced parasite prevalence. Sheep (90.6%) exhibited a significantly higher infection rate than cattle (52.0%) (OR = 8.95, 95% CI: 4.09–20.12, $P < 0.05$). Within the same host species, goats in East Ujimqin County (90.9%) had a significantly higher infection rate than those in Sonid Left Banner (40.0%) (OR = 15.00, 95% CI: 1.27–208.10, $P < 0.05$). Sheep in East Ujimqin County (95.6%) also had a significantly higher infection rate compared to those in Xilinhot (79.1%) (OR = 5.76, 95% CI: 1.61–17.56, $P < 0.05$).

Further analysis of co-infection with two or more parasites revealed an overall mixed infection rate of 47.8% (98/205). Among the livestock population, 30.7% (63/205) had two parasites, 15.1% (31/205) had three parasites, 1.5% (3/205) had four parasites, and 0.5% (1/205) were infected with five parasites (Table S1). The most common co-infection type was protozoan-helminth co-infection, detected in 38.0% (78/205) of animals, followed by protozoan-protozoan (8.9%, 18/205) and nematode-nematode (0.9%, 2/205) combinations. Livestock infected with *Eimeria* spp. showed a higher prevalence of other intestinal parasites (71.3%, 87/122) than non-infected animals (47.0%, 39/83). Statistical analysis indicated that *Eimeria* spp. infection was associated with an increased probability of co-infection with other intestinal parasites (OR = 2.80, 95% CI: 1.58–5.08, $P < 0.05$).

Overall, intestinal parasite co-infections in cattle and sheep from the Xilingol League were detected across all surveyed regions, showing clear regional and host-associated variation, with protozoan-helminth combinations being the most common type identified.

## DISCUSSION

Inner Mongolia, characterized by a semi-nomadic pastoral system that allows large-scale livestock movement, represents a critical ecological interface for zoonotic disease transmission in the northern grasslands of China (26). In regions such as the Xilingol League, where free-range grazing predominates, shared water sources and communal

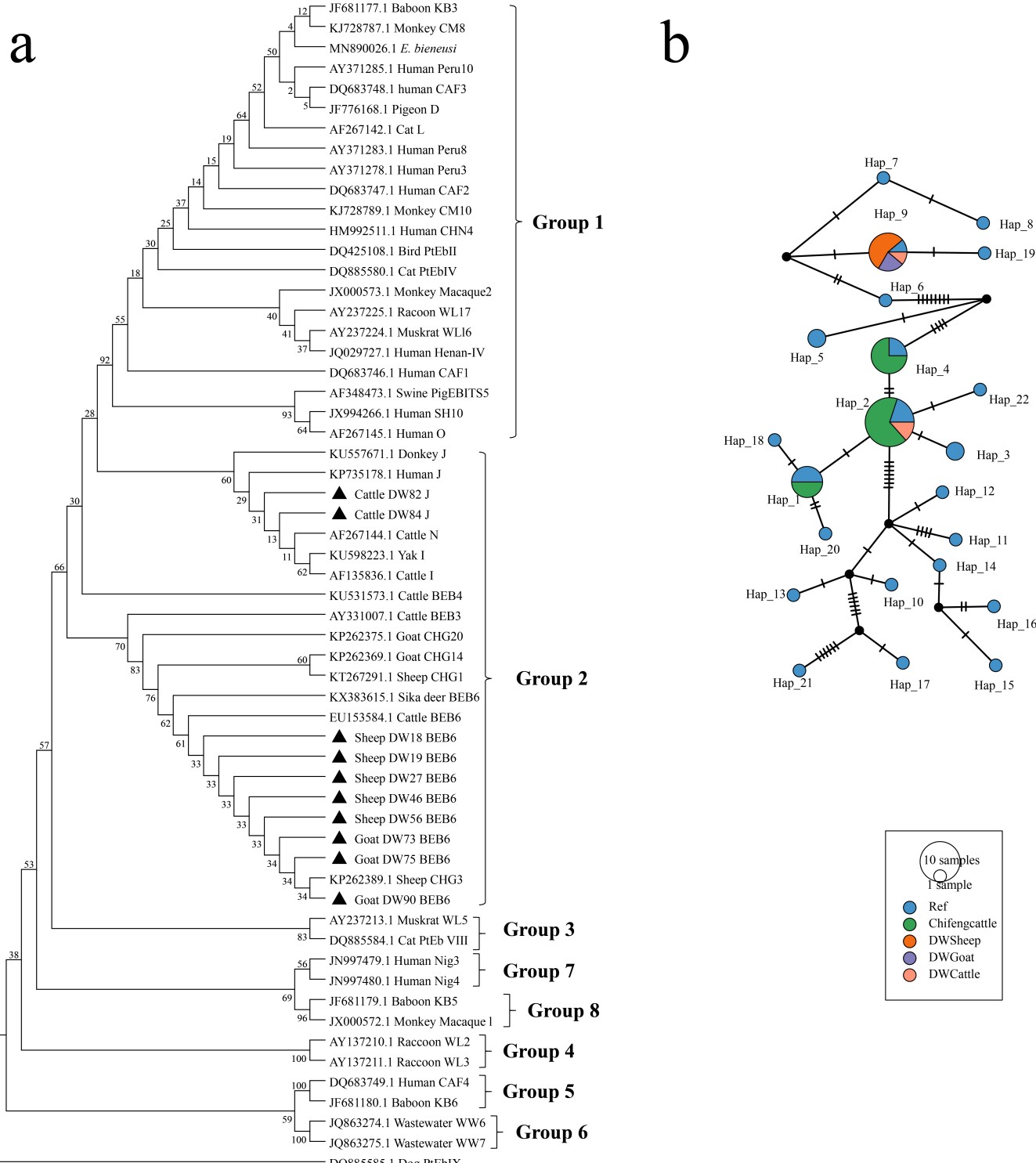

**FIG 4** Phylogenetic tree and haplotype analysis of *E. bieneusi*. (a) Phylogenetic tree of *E. bieneusi ITS* gene. (b) Haplotype analysis results of *E. bieneusi* sequences detected in this study and published *E. bieneusi* sequences. Different colors represent different groups, including Chifeng cattle, DW sheep, DW goat, and DW cattle, and the sizes of the circles represent the number of haplotypes.

pastures facilitate fecal contamination and cross-species transmission of intestinal parasites, which remain major constraints on livestock health and productivity. Despite their economic and zoonotic importance, intestinal parasitic infections have received limited attention in regional livestock health programs, and comprehensive data on

mixed infections are lacking (27, 28). To address this gap,the prevalence and genetic diversity of key intestinal parasites (*G. duodenalis*, *Cryptosporidium* spp., and *E. bieneusi*) in grazing livestock across Xilingol League was investigated, thereby establishing a baseline for understanding co-infection dynamics and informing integrated control strategies.

## Regional and host-specific variations in parasite prevalence

Significant regional and host-associated differences were observed in the prevalence of gastrointestinal parasites among livestock in the Xilingol League. *E. bieneusi* prevalence was substantially lower than values reported in Hulunbuir, Inner Mongolia, suggesting regional variation in parasite distribution (29). Compared with other regions of China, the overall prevalence of intestinal helminths (nematodes and trematodes) was relatively low, whereas *Eimeria* spp. and *G. duodenalis* exhibited notably higher infection rates (30). Marked spatial differences were also evident across the study area (31–33). Livestock from East Ujimqin County and Xilinhot showed consistently higher infection rates than those from Sonid Left Banner. These regional differences likely reflect local climatic and ecological conditions that influence parasite transmission, including precipitation, vegetation, and pasture contamination. Humid environments such as those in East Ujimqin County and Xilinhot may favor parasite survival and dissemination, whereas the arid conditions of Sonid Left Banner may constrain their persistence.

## Genetic variability of parasites across hosts and regions

Molecular analyses confirmed the presence of *G. duodenalis* assemblages B and E in cattle and sheep from Xilingol League, with assemblage B recognized as a zoonotic genotype (34). This finding suggests a potential public health risk associated with *G. duodenalis* infections in this region. For *Cryptosporidium* spp., only *C. andersoni* was detected in cattle, whereas sheep and goats were exclusively infected with *C. xiaoi*. Previous studies in Inner Mongolia have reported the presence of *C. parvum*, *C. andersoni*, *C. ryanae*, and *C. bovis* in cattle feces, and *C. ubiquitum*, *C. andersoni*, and *C. xiaoi* in sheep and goats (35, 36), indicating regional specificity in *Cryptosporidium* spp. distribution. Regarding *E. bieneusi*, genotypes BEB6 and J were identified as the predominant types in cattle and sheep, consistent with those previously described in ruminants (3). Positive samples were mainly found in East Ujimqin County, where Hap_9 was the most common haplotype, suggesting limited genotype diversity and localized distribution in this region.

## *Eimeria* spp. infection facilitates mixed intestinal parasitic infections

*Eimeria* infection was significantly associated with increased co-infection by other intestinal parasites in herbivores from Xilingol League, consistent with findings in ruminants elsewhere (37). Mixed infections are particularly common under free-grazing systems, where environmental exposure enhances transmission risk (37, 38). Mechanistically, *Eimeria* infection may promote secondary colonization by disrupting intestinal integrity and modulating host immunity (37). The resulting epithelial damage and inflammation can increase susceptibility to additional infections and maintain chronic intestinal disorders. Collectively, these findings suggest that *Eimeria* acts as a key predisposing factor for mixed parasitic infections, underscoring the need for integrated management strategies to improve livestock health and productivity.

This study revealed clear host- and region-associated patterns of intestinal parasitic infections in livestock from Xilingol League. *Eimeria* spp. were the most prevalent parasites, followed by helminths and intestinal protozoa, with sheep showing higher infection levels than cattle and goats. Molecular analyses demonstrated limited genetic variation but regionally shared haplotypes of *G. duodenalis* and *Cryptosporidium* spp. Mixed infections were common, and *Eimeria* infection notably increased the likelihood of co-infection with other intestinal parasites. Although species-level identification of *Eimeria*, nematodes, and trematodes was not performed, the findings provide a

comprehensive overview of parasite distribution and co-infection dynamics. These results fill an important epidemiological gap and support the need for integrated, species-targeted parasite control strategies in grazing livestock.

## MATERIALS AND METHODS

### Collection of fecal samples

Fecal samples were collected between 1 August and 5 August 2021, from three pastures in Xilingol League, Inner Mongolia: Sonid Left Banner (SZ), East Ujimqin County (DW), and Xilinhot (BY). A total of 205 fresh fecal samples were obtained, including 139 from sheep, 16 from goats, and 50 from cattle. Specifically, in Sonid Left Banner, samples were collected from 5 sheep, 5 goats, and 12 cattle; in East Ujimqin County, from 91 sheep, 11 goats, and 38 cattle; and in Xilinhot, from 43 sheep. All fecal samples were immediately stored at 4°C after collection and aliquoted for further analysis.

### Detection of parasite eggs using the saturated saline flotation method

The saturated saline flotation method was used for parasite egg or oocyst enrichment and identification in fecal samples, following a standardized protocol. Ten grams of fecal sample was accurately weighed and transferred into a 50 mL clean beaker. Twenty milliliters of saturated sodium chloride solution was then added and thoroughly stirred with a glass rod until homogenized. The mixture was then filtered through a 60-mesh stainless steel sieve, with the filtrate collected in a 50 mL centrifuge tube. The sample was centrifuged at $3,500 \times g$ for 5 min at 4°C. The upper layer was discarded, and the remaining liquid was gradually replaced with saturated sodium chloride solution until a convex liquid meniscus formed at the tube opening. Immediately, an 18 mm × 18 mm coverslip was placed on the liquid surface. After 2 min, the coverslip was carefully lifted vertically and placed onto a glass slide. Parasite eggs were observed under an optical microscope. The identification of egg morphology was based on standard parasitology atlases (39).

### Extraction of fecal DNA

Genomic DNA was extracted from fecal samples using the FastDNA Stool Kit (MP Biomedicals, Irvine, CA, USA). Approximately 500 mg of fecal sample was processed in accordance with the manufacturer's protocol (35). Extracted DNA samples were stored at −20°C for further analysis.

### Molecular identification of intestinal protozoa using multiplex nested PCR

Because protozoan cysts are small (2 μm–5 μm in diameter), the saturated salt flotation method showed limited sensitivity for detection (27, 28). To improve diagnostic accuracy and enable genotyping, nested PCR was used for molecular identification of *G. duodenalis*, *Cryptosporidium* spp., and *E. bieneusi*, targeting the bg, SSU rRNA, and the ITS loci, respectively (40–42). The primer sequences, annealing temperatures, and expected amplicon sizes are provided in Table 2. Each PCR batch incorporated both a positive control (previously confirmed positive samples) and a negative control (sterile ultrapure water). The second-round PCR products (5 μL) were subjected to 1.0% (wt/vol) agarose gel electrophoresis, stained with GelstainRed (UElandy, Suzhou, Jiangsu, China), and visualized under a UV transilluminator. Positive amplicons were sent to Sangon Biotech Co., Ltd. (Shanghai, China) for bidirectional sequencing using the ABI PRISM 3730 XL DNA Analyzer (Applied Biosystems, USA). The quality of the sequence was then assessed using Chromas software, and the sequence alignment was performed via BLAST (https://blast.ncbi.nlm.nih.gov/Blast.cgi). The genotype of *E. bieneusi* was determined based on established nomenclature systems (43), while *Cryptosporidium* spp. and *G. duodenalis* genotyping followed national guidelines (36).

**TABLE 2** The information of nested PCR primers of three protozoa detection

| Parasite species | Gene locus | Primer sequences （5′→3′) | Fragment size/bp | Annealing temperature/℃ |
|---|---|---|---|---|
| *Cryptosporidium* spp. | *SSU rRNA* | F1: GAC ATA TCA TTC AAG TTT CTG ACC R1: CTG AAG GAG TAA GGA ACA ACC F2: CCT ATC AGC TTT AGA CGG TAG G R2: TCT AAG AAT TTC ACC TCT GAC TG | 587 | 58 58 |
| *E. bieneusi* | *ITS* | F1: GGT CAT AGG GAT GAA GAG R1: TTC GAG TTC TTT CGC GCT C F2: GCT CTG AAT ATC TAT GGC T R2: ATC GCC GAC GGA TCC AAG TG | 392 | 57 55 |
| *G. duodenalis* | *bg* | F1: AAG CCC GAC GAC CTC ACC CGC AGT GC R1: GAG GCC GCC CTG GAT CTT CGA GAC GAC F2: GAA CGA ACG AGA TCG AGG TCC G R2: CTC GAC GAG CTT CGT GTT | 511 | 65 55 |

## Haplotype network and phylogenetic analysis

A neighbor-joining phylogenetic tree was constructed to analyze the genetic structure of intestinal parasites in Xilingol League. Reference sequences were downloaded from GenBank to form a comparative data set, including *Cryptosporidium* spp. (*C. xiaoi*, *C. bovis*, *C. ryanae*, *C. andersoni*, *C. canis*, *C. ubiquitum*, *C. suis*, *C. parvum*, and *C. felis*) based on *SSU rRNA* gene sequences; *G. duodenalis* assemblages A, B, C, D, and E based on *bg* gene sequences; *E. bieneusi* based on *ITS* gene sequences. Multiple sequence alignment and phylogenetic analyses were performed using MEGA 11 software (version 11.0.13). Genetic distances were estimated using the Kimura two-parameter model, followed by neighbor-joining phylogenetic tree construction. Bootstrap analysis with 1,000 replicates was conducted to assess the robustness of tree topology (44). Outgroup sequences were selected as follows: *C. felis* (MH115431.1) for *Cryptosporidium* spp., *G. duodenalis* assemblage C (LC437439.1) for *G. duodenalis*, and *E. bieneusi* genotype PtEbIX (DQ885585.1) for *E. bieneusi*.

To achieve further genetic characterization of local strains, reference sequences were retrieved from GenBank, including the *C. xiaoi SSU rRNA* gene, *G. duodenalis* assemblage B *bg* gene, and *E. bieneusi ITS* gene sequences. Haplotype identification was performed using DnaSP software (version 6.12.03). Haplotype network analysis was conducted using the TCS method with a 95% connection limit (45). The resulting haplotype network was visualized in PopART (version 1.7), with sample frequency annotations for each identified haplotype (46).

## Statistical analysis

Differences in intestinal parasite prevalence between groups were assessed using the $\chi^2$ test. ORs and 95% CIs were calculated to evaluate the association between risk factors and parasitic infections. All statistical analyses were performed using SPSS software (version 25.0). Statistical significance was set at $P < 0.05$.

## ACKNOWLEDGMENTS

This work was supported by the Study on Key Technology Project of the Inner Mongolia Science and Technology Department (2021GG0171); pathogen spectrum, temporal and spatial distribution, and transmission features of the important emerging and re-emerging zoonosis in Inner Mongolia Autonomous Region (U22A20526); Inner Mongolia Autonomous Region Science and Technology Leading Talent Team: Zoonotic Disease Prevention and Control Technology Innovation Team (2022SLJRC0023); and the Young Scientists Fund of the National Natural Science Foundation of China (32402911).

W.H. and W.Y. conceived the study and contributed the original idea. Z.M., Junyan Li, and J.Q. performed the experiments. Z.M. and J.Q. wrote the initial draft of the paper. Junyan Li, W.Y., and W.H. contributed to the revision of the manuscript, and the final version was reviewed by W.H. All authors approved the final manuscript.

## AUTHOR AFFILIATIONS

[1]College of Life Sciences, Inner Mongolia University, Hohhot, China

[2]Veterinary Research Institute, Inner Mongolia Academy of Agricultural and Animal Husbandry Sciences, Hohhot, China

[3]Inner Mongolia Key Laboratory of Tick-borne Zoonotic Infectious Disease, Department of Medicine, Hetao College, Bayannur, China

[4]Institutes of Biomedical Sciences, Inner Mongolia University, Hohhot, China

[5]Key Laboratory of Parasite and Vector Biology of China Ministry of Health, WHO Collaborating Centre for Tropical Diseases, Joint Research Laboratory of Genetics and Ecology on Parasite-Host Interaction, Fudan University, National Institute of Parasitic Diseases, Chinese Center for Disease Control and Prevention, Shanghai, China

[6]Department of Infectious Diseases, State Key Laboratory of Genetic Engineering, Ministry of Education Key Laboratory for Biodiversity Science and Ecological Engineering, Ministry of Education Key Laboratory of Contemporary Anthropology, School of Life Sciences, Fudan University, Huashan Hospital, Shanghai, China

## AUTHOR ORCIDs

Ziran Mo  http://orcid.org/0009-0008-2276-186X
Wenbin Yang  http://orcid.org/0009-0003-5392-221X
Junyan Li  http://orcid.org/0009-0009-9952-6339
Wei Hu  http://orcid.org/0000-0002-4432-5400

## FUNDING

| Funder | Grant(s) | Author(s) |
|---|---|---|
| Department of Science and Technology of Inner Mongolia Autonomous Region | 2021GG0171 | Wei Hu |
| Inner Mongolia University | U22A20526 | Wei Hu |
| Department of Science and Technology of Inner Mongolia Autonomous Region | 2022SLJRC0023 | Wei Hu |
| National Natural Science Foundation of China | 32402911 | Wenbin Yang |

## AUTHOR CONTRIBUTIONS

Ziran Mo, Conceptualization, Methodology, Visualization, Writing – original draft | Jingwei Quan, Methodology, Writing – original draft | Jiahao Dao, Methodology | Xiaoping Luo, Methodology | Penglong Wang, Methodology | Yurun Miao, Methodology | Lei Xiu, Methodology | Gaowa Gong, Methodology | Jian Li, Methodology | Wenbin Yang, Conceptualization, Writing – original draft, Writing – review and editing | Junyan Li, Conceptualization, Writing – original draft | Wei Hu, Conceptualization, Funding acquisition, Writing – original draft, Writing – review and editing

## DATA AVAILABILITY

The representative sequences obtained in this study have been deposited in GenBank with the following accession numbers: *G. duodenalis*: OQ735297–OQ735300, *Cryptosporidium* spp.: OQ727294–OQ727297 and *E. bieneusi*: OQ732700–OQ732703.

## ETHICS APPROVAL

Animal sample collection was conducted with the consent of the farm owners.

## ADDITIONAL FILES

The following material is available online.

## Supplemental Material

**Table S1 (Spectrum01697-25-s0001.xlsx).** Summary of sample information and test results.

## Open Peer Review

**PEER REVIEW HISTORY (review-history.pdf).** An accounting of the reviewer comments and feedback.

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
