## [Reviewer comments · Microbiology Spectrum]

Microbiology Spectrum

Prevalence and genetic diversity of intestinal parasites in Xilingol League herbivores, Inner Mongolia, northern China

Mo Ziran, Quan Jingwei, Dao Jiahao, Luo Xiaoping, Wang Penglong, Miao Yurun, Xiu Lei, Gaowa gong, Li Jian, Yang Wenbin, Li Junyan, and Hu Wei

Corresponding Author(s): Mo Ziran, Inner Mongolia University

Review Timeline:

Submission Date:	June 2, 2025
Editorial Decision:	August 27, 2025
Revision Received:	October 22, 2025
Accepted:	November 24, 2025

Editor: Jian Li

Reviewer(s): Disclosure of reviewer identity is with reference to reviewer comments included in decision letter(s). The following individuals involved in review of your submission have agreed to reveal their identity: Si-Yang Huang (Reviewer #1)

Transaction Report:

DOI: <https://doi.org/10.1128/spectrum.01697-25>

Re: Spectrum01697-25 (**Intestinal parasite in Xilingol livestock: prevalence and genetic diversity**)

Dear Dr. Mo Ziran:

Thank you for the privilege of reviewing your work. Below you will find my comments, instructions from the Spectrum editorial office, and the reviewer comments.

Revision Guidelines

Sincerely,
Jian Li
Editor
Microbiology Spectrum

Reviewer #1 (Comments for the Author):

1. The title should be changed to "Intestinal parasite in Xilingol herbivore: prevalence and genetic diversity"
2. The sample size quite small in Xilinhot and Sonid Left Banner, the author should explain the reason, the total number is less or some other reason?
3. cattle, sheep, and goats should be calculated respectively.

Reviewer #2 (Comments for the Author):

The present study investigated the prevalence and genetic diversity of intestinal parasites in livestock from Xilinggol of Inner Mongolia, northern China. It is well-designed and performed, and the findings provide basic data for controlling infection of these parasites in this region. However, the language should be improved, and just several suggestions are listed as following:

Title

Suggest to be "Prevalence and genetic diversity of intestinal parasites in livestock from Xilinggol of Inner Mongolia, northern China".

Abstract

Line 37, change "livestock productivity" to be "productivity of livestock".

Line 45-47, this sentence should be rephrased.

Line 47, the word *Eimeria* should be italic.

The species of *Eimeria* should be identified, and the information should be provided in the abstract.

Importance

Line 59-61, this sentence should be rephrased.

Introduction

Line 92-94, this sentence should be rephrased.

Line 92-99, these findings should be in discussion.

Lines 67-128, The Introduction is somewhat verbose, with partially repetitive content between the first and second paragraphs. A more concise structure is suggested-e.g., consolidating global context and narrowing down to the regional gap-to better highlight the rationale of the study.

Results

Lines 147-149, The sentence "with the highest prevalence observed in sheep. Moreover, East Ujimqin County and Xilinhot exhibited markedly higher infection rates compared to Sonid Left Banner." contains redundant expression. Please revise to avoid repetition.

Line 170, what species of *Eimeria*, trematodes and nematodes?

Line 188, how about *tpi* and *gdh* loci?

Lines 280-282, Replace "protozoa-nematode" with the correct terminology: "protozoan-helminth co-infection". The terms "mixed infection," "co-infection," and "concurrent infection" are used interchangeably. Please standardize terminology throughout the manuscript to improve clarity, preferably using "co-infection" as the primary term.

Discussion

Line 306, The sentence "are often overlooked in routine livestock management, and existing research has focused..." contains mismatched components. It is suggested to split or rephrase the sentence to improve clarity and readability.

Line 321, The statement referring to a 58.3% prevalence of intestinal helminths in other regions of China lacks citation. Please provide the corresponding reference to support this comparative figure.

Line 370, "*Eimeria*" should be italicized.

Methods

Line 388-389, give the detail information of sample collection.

Reviewer #1 (Comments for the Author):

Response: Dear Reviewer #1, We sincerely thank you for your insightful and constructive comments on our manuscript. We are very grateful for your efforts in reviewing our work. In revising the manuscript, we carefully addressed each of your concerns. We also apologize for the writing and content errors that were originally overlooked and have now been corrected in accordance with your suggestions. Below is a detailed point-by-point response.

1. The title should be changed to "Intestinal parasite in Xilingol herbivore: prevalence and genetic diversity"

Response: Thank you for your suggestion. We agree that using “herbivore” can emphasize the host type. After carefully considering both your advice and Reviewer #2’ s recommendation, we have revised the title to:

“Prevalence and genetic diversity of intestinal parasites in Xilingol League herbivores, Inner Mongolia, northern China.”

2. The sample size quite small in Xilinhot and Sonid Left Banner, the author should explain the reason, the total number is less or some other reason?

Response:

Thank you for pointing this out. The smaller sample sizes in Xilinhot (n = 43) and Sonid Left Banner (n = 22) were due to the following reasons:

Livestock availability: Herd sizes and the number of eligible farms/households in these two regions were considerably lower than in East Ujimqin County, which limited the pool of animals that could be sampled.

Proportional sampling: To maintain representativeness, we applied proportional sampling relative to the local herd census, which naturally resulted in fewer samples from areas with smaller livestock populations.

Logistical constraints: In addition, herds in these two counties were more scattered, and access to some farms was restricted, further reducing the number of samples collected.

Despite these limitations, all samples were collected randomly within each site, ensuring that the data adequately represent the local livestock populations.

3. Cattle, sheep, and goats should be calculated respectively.

Response: Thank you for this valuable suggestion. We fully agree with the importance

of reporting host-specific infection data. In the revised manuscript:

Abstract update: Infection rates for cattle, sheep, and goats are now explicitly presented in the Abstract to provide a clearer overview of host-specific differences.

(Lines 45-46)

Results revision: We have restructured the Results section to present prevalence and statistical comparisons separately for each host species. (Lines 119-121, 134-136, 153-156, 200-202, 227-229)

Reviewer #2 (Comments for the Author):

The present study investigated the prevalence and genetic diversity of intestinal parasites in livestock from Xilingol of Inner Mongolia, northern China. It is well-designed and performed, and the findings provide basic data for controlling infection of these parasites in this region. However, the language should be improved, and just several suggestions are listed as following:

Response: Dear Reviewer #2:

Thank you for your positive evaluation and constructive suggestions. We carefully considered your comments, corrected the inconsistencies, and revised the manuscript to improve clarity and scientific quality. Detailed responses are provided below.

Title

1. Suggest to be "Prevalence and genetic diversity of intestinal parasites in livestock from Xilingol of Inner Mongolia, northern China".

Response: We appreciate your constructive suggestion. Following your advice, we restructured the title for clarity. At the same time, considering Reviewer #1's comment, we replaced "livestock" with "herbivores" to better describe the study population. The final revised title is:

"Prevalence and genetic diversity of intestinal parasites in Xilingol League herbivores, Inner Mongolia, northern China."

Abstract

2. Line 37, change "livestock productivity" to be "productivity of livestock".

Response: Thank you for the suggestion. We have revised the wording from "livestock productivity" to "productivity of livestock" and harmonized this phrasing throughout the

manuscript where applicable. (Line 38)

3. Line 45-47, this sentence should be rephrased.

Response: Thank you for pointing this out. We have rephrased the sentence in Lines 45 – 47 to improve clarity and readability. The revised version now reads:

(Genetic characterization further revealed distinct dominant lineages, including *G. duodenalis* (assemblages B and E), *Cryptosporidium* (*C. andersoni* and *C. xiaoi*), and *E. bienersi* (genotypes J and BEB6).) (Line 48-50)

4. Line 47, the word *Eimeria* should be italic.

Response: Thank you for pointing this out. We have corrected the formatting so that *Eimeria* appears in italics. We also checked the entire manuscript to ensure all Latin genus/species names are consistently italicized. (Line 58)

5. The species of *Eimeria* should be identified, and the information should be provided in the abstract.

Response: We thank the reviewer for this valuable suggestion. We agree that species-level identification of *Eimeria* would substantially enhance the epidemiological value of the study. However, our current diagnostic workflow focused on estimating overall prevalence and did not include species-level typing, so we are unable to provide specific species information. To avoid overinterpretation, we have kept the results at the genus level and added a note in the Discussion acknowledging this limitation. We also outline plans to incorporate species determination (morphology plus targeted molecular typing) in future investigations.

6. Line 59-61, this sentence should be rephrased.

Response: Thank you for your thoughtful comment. Since our diagnostic workflow was centered on prevalence estimation, species-level identification of *Eimeria* was not undertaken in this study. We acknowledge this limitation and plan to incorporate molecular typing in future work. To address this, we have revised the sentence to more accurately reflect the statistical association without suggesting causality. (Line 64-66)

Introduction

7. Line 92-94, this sentence should be rephrased.

Response: Thank you for pointing this out. We have rephrased the sentence at Lines 95 – 97

to improve clarity and readability. The revised version now reads:

(Intestinal parasitic infections frequently occur as co-infections involving multiple parasite species, which can exacerbate pathological outcomes and increase transmission potential)

8. Line 92-99, these findings should be in discussion.

Response: Thank you for the suggestion. We have moved the content from Lines 92 – 99 to the Discussion to improve logical flow.

9. Lines 67-128, The Introduction is somewhat verbose, with partially repetitive content between the first and second paragraphs. A more concise structure is suggested-e.g., consolidating global context and narrowing down to the regional gap-to better highlight the rationale of the study.

Response: Thank you for this constructive suggestion. We have streamlined the Introduction by removing redundant content and consolidating the first two paragraphs. The revised structure now provides a concise global context followed by a clear statement of the regional knowledge gap and study rationale.

Results

10. Lines 147-149, The sentence "with the highest prevalence observed in sheep. Moreover, East Ujimqin County and Xilinhot exhibited markedly higher infection rates compared to Sonid Left Banner." contains redundant expression. Please revise to avoid repetition.

Response: Thank you for pointing this out. We have removed the redundant wording and consolidated the passage into a single concise statement:

“Overall, *Eimeria* spp. infection exhibited distinct regional and host-associated differences, with the highest prevalence recorded in sheep and in samples collected from East Ujimqin County and Xilinhot.” (line124-126)

11. Line 170, what species of Eimeria, trematodes and nematodes?

Response: We thank the reviewer for this important comment. While experienced parasitologists may sometimes infer likely species based on egg or oocyst morphology, these features often overlap among closely related species and do not allow unequivocal identification. To ensure accuracy and avoid potential misclassification, we reported these parasites at the genus level. We have also added a note in the Discussion acknowledging this

limitation and highlighting that species-level identification will require integrated morphological and molecular approaches in future work. (line 311-314)

12. Line 188, how about tpi and gdh loci?

Response: Thank you for this valuable suggestion. The tpi and gdh loci, together with bg, are commonly used markers for *Giardia duodenalis* genotyping, with multilocus sequence typing providing higher resolution for sub-assemblage differentiation. In the present study, we focused on the bg locus, which is widely used in prevalence surveys and provides reliable genotyping results consistent with standard practice. We acknowledge the value of including tpi and gdh in future work to enhance discriminatory power, but our current approach remains consistent with previous epidemiological studies.

- **Sursal N, Simsek E, Yildiz K. Feline Giardiasis in Turkey: Prevalence and genetic and haplotype diversity of *Giardia duodenalis* based on the β -Giardin gene sequence in symptomatic cats[J]. *The Journal of Parasitology*, 2020, 106(5): 699-706.**
- **de Lucio A, Amor-Aramendía A, Bailo B, et al. Prevalence and genetic diversity of *Giardia duodenalis* and *Cryptosporidium* spp. among school children in a rural area of the Amhara Region, North-West Ethiopia[J]. *PloS one*, 2016, 11(7): e0159992.**

13. Lines 280-282, Replace "protozoa-nematode" with the correct terminology: "protozoan-helminth co-infection". The terms "mixed infection," "co-infection," and "concurrent infection" are used interchangeably. Please standardize terminology throughout the manuscript to improve clarity, preferably using "co-infection" as the primary term.

Response: Thank you for the correction and helpful suggestion. We have revised the text at Lines 280 – 282 to read “protozoan-helminth co-infection.” In addition, we have standardized terminology throughout the manuscript by adopting “co-infection” as the primary term, replacing instances of “mixed infection” and “concurrent infection” for clarity and consistency.

Discussion

14. Line 306, The sentence "are often overlooked in routine livestock

management, and existing research has focused..." contains mismatched components. It is suggested to split or rephrase the sentence to improve clarity and readability.

Response: Thank you for pointing this out. We have revised the sentence at Line 306 to improve clarity. The revised version now reads:

“However, intestinal parasitic infections are often overlooked in routine livestock management. Most existing studies have focused on the detection of single-parasite infections, while systematic evaluations of mixed infections remain largely unexplored.” (Line 323-326)

15. Line 321, The statement referring to a 58.3% prevalence of intestinal helminths in other regions of China lacks citation. Please provide the corresponding reference to support this comparative figure.

Response: Thank you for your careful reading and valuable comment. We have revised the statement at line 321 and removed the unsupported figure. The revised text no longer contains the 58.3% prevalence rate, and therefore the issue of missing citation has been resolved.

16. Line 370, "*Eimeria*" should be italicized.

Response: Thank you for catching this. We have corrected the formatting so that *Eimeria* appears in italics at that occurrence. (Line 390)

Methods

17. Line 388-389, give the detail information of sample collection.

Response: Thank you for this helpful comment. We have now revised the manuscript to provide more detailed information regarding the sample collection, including the number and species of animals sampled from each location. (Line 416-418)

Re: Spectrum01697-25R1 (**Prevalence and genetic diversity of intestinal parasites in Xilingol League herbivores, Inner Mongolia, northern China**)

Dear Dr. Mo Ziran:

Your manuscript has been accepted, and I am forwarding it to the ASM production staff for publication. Your paper will first be checked to make sure all elements meet the technical requirements. ASM staff will contact you if anything needs to be revised before copyediting and production can begin. Otherwise, you will be notified when your proofs are ready to be viewed.

Sincerely,
Jian Li
Editor
Microbiology Spectrum

Reviewer #1 (Comments for the Author):

no

Reviewer #2 (Comments for the Author):

The authors addressed all points raised by the Reviewers.